# Metabolites with Anti-Inflammatory Activities Isolated from the Mangrove Endophytic Fungus *Dothiorella* sp. ZJQQYZ-1

**DOI:** 10.3390/microorganisms13040890

**Published:** 2025-04-12

**Authors:** Zhaokun Li, Junhao Zhu, Ruxue Mu, Chenxi Wang, Yuru Sun, Bingbing Qian, Ning Li, Yan Chen

**Affiliations:** School of Pharmacy, Anhui Medical University, Hefei 230032, China

**Keywords:** mangrove endophytic fungus, *Dothiorella* sp., anti-inflammatory

## Abstract

As special condition-derived microorganisms, mangrove endophytic fungi can produce abundant and active secondary metabolites. In this paper, one strain of *Dothiorella* sp. ZJQQYZ-1 was isolated from *Kandelia candel*. As a result, six compounds were obtained from *Dothiorella* sp. ZJQQYZ-1, including three new benzofuran derivatives dothiofurans A-C (**1**–**3**), one new sesquiterpene dothiopene A (**4**), one new steroid phomosterol C (**5**), and three known analogs phomosterol B (**6**), phomosterol A (**7**), and pergillin (**8**). Their structures were characterized by extensive spectroscopic analysis, electronic circular dichroism (ECD), and ^13^C NMR calculations. The bioactive assay showed that **7** exhibited significant anti-inflammatory activity with an IC_50_ value of 4.6 μM. Furthermore, **7** effectivity suppressed the protein expression of inducible nitric oxide synthase (iNOS) in LPS-stimulated RAW264.7 cells.

## 1. Introduction

*Dothiorella*, with a wide range of hosts such as endophytes, saprobes, and pathogens, is a cosmopolitan species [1]. The secondary metabolites from *Dothiorella* exhibited diverse biological activities, like cytotoxicity [2] and antimicrobial [3]. Mangrove endophytic fungi grow inside healthy plant tissues of mangroves. They are symbiotic with the host plant and can improve plant disease resistance and growth rate. Secondary metabolites derived from mangrove endophytic fungi are structurally novel, and have a wide range of biological activities, like anti-inflammatory [4], antimicrobial [5], anticancer [6,7], acetylcholinesterase inhibitory [8], antibacterial [9], cytotoxic [10,11], α-glucosidase inhibitory [12,13], and antiviral [14].

Inflammation involving the innate and adaptive immune systems is a normal response to infection, but it could lead to autoimmune or autoinflammatory disorders, neurodegenerative disease, or cancer when allowed to continue without prevention [15]. Nowadays, the anti-inflammatory drugs commonly used in clinical practice include glucocorticoids and non-steroidal drugs, but they all have side effects, such as drug resistance and allergic reactions [16]. New anti-inflammatory drugs with non-toxicity need to be discovered. Natural medicines have been investigated extensively in recent years for fewer adverse effects and multiple targets. It is the resource of anti-inflammatory drugs [17]. As is well known that more than half of the drugs have been directly or indirectly derived from natural products from 1981 to 2019 [18]. Like alkaloids [19], terpenoids [20], polyketones [19], steroidal saponins [21], and polysaccharides [22] showed unexpected anti-inflammatory activities.

In our continuous research into the compounds with anti-inflammatory activities from mangrove endophytic fungi, the strain *Dothiorella* sp. ZJQQYZ-1 from *Kandelia candel* attracted our attention through the guidance of biological activity. As a result, three new benzofuran derivatives dothiofurans A-C (**1**–**3**), one new sesquiterpene dothiopene A (**4**), one new steroid phomosterol C (**5**), and three known analogues phomosterol B (**6**) [23], phomosterol A (**7**) [23], and pergillin (**8**) [24] (Figure 1) were isolated from the mangrove endophytic fungus *Dothiorella* sp. ZJQQYZ-1. The anti-inflammatory potential of all the compounds was evaluated on macrophages. Compounds **7** and **5** showed significant NO inhibitory activity with IC_50_ values of 4.6 and 29.3 µM, respectively. Herein, we report the isolation, structural determinations, and biological evaluation of these isolated compounds.

## 2. Materials and Methods

### 2.1. General Experimental Procedures

The experiment used the same instruments as a previously published paper [23]. Briefly, the UV data were obtained by an Evolution 200 spectrophotometer (Thermo Fisher, Waltham, MA, USA). The electronic circular dichroism (ECD) spectra were recorded on a Model 420SF (Aviv biomedical Inc., Lakewood, NJ, USA). The IR spectra of new compounds were measured using the Nicolet iS 10 FT-IR spectrometer with KBr disks (Thermo Fisher). ^1^H and ^13^C NMR and 2D NMR spectra were detected on a Bruker Avance 500 spectrometer using TMS as an internal standard (Bellerica, MA, USA). The high-resolution electrospray ionization mass spectra (HRESIMS) were collected using a Q-TOF4600 mass spectrometer (AB SCIEX, Waltham, MA, USA). Silica gel (200–300 mesh, Qingdao Marine Chemical Factory, Qingdao, Shandong, China) and Sephadex LH-20 (Amersham Biosciences, Little Chalfont, UK) were used for column chromatography (CC). Thin-layer chromatography (TLC) was conducted on silica gel plates using HSGF254 from Yantai HuangHai Silica company (Yantai, Shandong, China).

### 2.2. Fungal Material

*Dothiorella* sp. ZJQQYZ-1 was isolated from the healthy leaves of *Kandelia candel* collected from Dongzhai Harbor Mangrove Nature Reserve in Hainan province of China. The ITS sequence of this strain showed 99% similarity to that of *Dothiorella* sp. (GenBank No. PQ559842). The voucher specimen has been preserved in the School of Pharmacy, Anhui Medical University.

### 2.3. Fermentation, Extraction and Isolation

The fungus *Dothiorella* sp. was first cultivated on potato dextrose medium (four 1 L Erlenmeyer flasks each containing 300 mL medium) to prepare the seed cultures. Then, the spore suspension was transferred into rice medium (120 × 1 L Erlenmeyer flasks each containing 150 g of rice and 70 mL of 0.3% sea-salt solution) for 30 days at 25 °C. After fermentation, the rice substrate was extracted four times with methyl alcohol. Then, a total extract of 408 g was obtained after removing the liquids in a vacuum. The extract was then subjected to a silica gel (200–300 mesh) column with a stepwise gradient elution of petroleum ether (PE) and EtOAc (*v*/*v*, 100:0, 90:10, 80:20, 70:30, 60:40, 50:50, 40:60, 30:70, and 0:100) to give ten fractions (Fr.1–Fr.10). Fr.4 was subjected to Sephadex LH-20 (CH_2_Cl_2_/MeOH,1:1 *v*/*v*) to afford three subfractions (Fr.4.1, Fr.4.2, and Fr.4.3). Compound **5** (1.8 mg) was yielded from Fr.4.1, which was purified on silica gel eluting with CH_2_Cl_2_/MeOH (150:1 to 100:2, *v*/*v*). Fr.4.2 was separated on silica gel eluting with CH_2_Cl_2_/MeOH (100:1 to 100:2, *v*/*v*) to obtain compound **4** (2.5 mg). Fr.4.3 was separated by column chromatography on silica gel eluting with CH_2_Cl_2_/MeOH (100:2 to 100:5, *v*/*v*) to obtain compounds **7** (3.5 mg) and **8** (4.3 mg). Fr.5 was purified by silica gel CC eluting with CH_2_Cl_2_-MeOH (100:1 to 100:4, *v*/*v*) to yield two subfractions (Fr.5.1–Fr.5.2). The two fractions were further separated by Sephadex LH-20 (CH_2_Cl_2_-MeOH, 1:1 *v*/*v*) to give compounds **1** (2.2 mg) and **2** (1.5 mg). Then, Fr.6.1 was obtained from Fr.6 by Sephadex LH-20 (CH_2_Cl_2_-MeOH, 1:1 *v*/*v*), and it was further subjected by silica gel CC eluting with CH_2_Cl_2_-MeOH (100:3 to 100:5, *v*/*v*) to obtain compound **3** (2.6 mg) and **6** (2.0 mg).

Dothiofuran A (**1**): white solid; ${\left[a\right]}_{D}^{25}$ = −37.0 (*c* = 0.24, MeOH); IR (KBr) *υ*_max_: 3381, 1690, 1652, 1511, 1450, 1382, 1266, 1158, 1000 cm^−1^; UV (MeOH) *λ*_max_: 347 (1.2), 283 (1.4), 277 (1.5), 208 (3.3) nm; HRESIMS: *m*/*z* 273.07687 [M − H]^−^ (calcd for C_15_H_14_O_5_, 273.07685); for ^1^H and ^13^C NMR (DMSO-*d*_6_) data see Table 1.

Dothiofuran B (**2**): yellow oil; ${\left[a\right]}_{D}^{25}$ = −15.6 (*c* = 0.14, MeOH); IR (KBr) *υ*_max_: 3380, 1690, 1656, 1508, 1448, 1380, 1265, 1155, 1012 cm^−1^; UV (MeOH) *λ*_max_: 347 (1.1), 282 (1.4), 276 (1.6), 209 (3.2) nm; HRESIMS: *m*/*z* 273.077.03 [M − H]^−^ (calcd for C_15_H_14_O_5_, 273.07685); for ^1^H and ^13^C NMR (CDCl_3_ and CD_3_OD) data see Table 1.

Dothiofuran C (**3**): white solid; ${\left[a\right]}_{D}^{25}$ = −10.2 (*c* = 0.11, MeOH); IR (KBr) *υ*_max_: 3381, 3200, 2925, 1650, 1486, 1382, 1262, 1068 cm^−1^; UV (MeOH) *λ*_max_: 294 (1.3), 216 (3.6) nm; HRESIMS: *m*/*z* 263.1277 [M − H]^−^ (calcd for C_15_H_20_O_4_, 263.1289); for ^1^H and ^13^C NMR (CDCl_3_) data see Table 1.

Dothiopene A (**4**): colorless oil; ${\left[a\right]}_{D}^{25}$ = +11.6 (*c* = 0.2, MeOH); IR (KBr) *υ*_max_: 3365, 1868, 1650, 1480, 1381, 1260, 1002 cm^−1^; UV (MeOH) *λ*_max_: 224 (2.0) nm; HRESIMS: *m*/*z* 273.14586 [M + Na]^+^ (calcd for C_15_H_22_NaO_3_, 273.14581); for ^1^H and ^13^C NMR (CDCl_3_) data see Table 1.

Phomosterol C (**5**): white solid; ${\left[\alpha \right]}_{D}^{25}$ = −16.8 (*c* 0.15, MeOH); UV (MeOH) *λ*_max_ (log *ε*): 215 (3.61) nm; IR (KBr) *ν*_max_: 3345, 3012, 2950, 1681, 1462, 1428, 1330, 1251, 1220, 1100 cm^−1^; for ^1^H and ^13^C NMR (500 MH*z*, CD_3_OD) data see Table 2; HRESIMS *m*/*z* 451.28097 [M + Na]^+^ (calcd for C_27_H_40_NaO_4_, 451.28090).

### 2.4. Inhibitory Activity of NO Production Activity Assay

#### 2.4.1. Cell Culture and Treatment

The RAW264.7 macrophage cell was obtained from Servicebio (Wuhan, Hubei, China) and was verified to be free of mycoplasma infection. It was cultured in Eagle medium Dulbecco’s modified eagle medium (DMEM, Sigma, St. Louis, MO, USA) supplemented with 10% fetal bovine serum (FBS, Royace, Lanzhou, Gansu, China) and 1% penicillin–streptomycin (Biosharp, Hefei, Anhui, China) in a humidified atmosphere with 5% CO_2_ at 37 °C. When the cells were 80% grown, they were passaged 1:3 or 1:4.

#### 2.4.2. Cell Viability Assay

The in vitro cell viability assay was conducted as what has been reported [25]. In a nutshell, when cells grew logarithmically, the cells were seeded in 96-well plates (Flat bottom, Labselect, Hefei, Anhui, China, 2 × 10^5^ cells/well, 100 μL/well). After incubation for 24 h (5% CO_2_, 37 °C), we aspirated the original medium and added a new medium (200 μL/well) with/without the various concentrations of compound treatment (3.125–50 μM). After incubation for 24 h again, the cells were treated with the CCK8 solution (Apexbio, Houston, TX, USA) with 110 μL/well for 1h. The assay kit utilizes a highly water-soluble tetrazolium salt, WST-8, which is reduced in the presence of electron coupling reagents to generate water-soluble formazan dye. The amount of formazan produced by dehydrogenases demonstrates a direct linear relationship with the number of viable cells. Then, absorbance was measured at 450 nm using a microplate reader (Thermo Fisher). The formula (Cell viability = (ODcompound − ODblank)/(ODCon − ODblank) × 100%) has been used to calculate the cell viability.

#### 2.4.3. In Vitro Measurement of Nitric Oxide Production

The in vitro NO production inhibitory assay was conducted as has been reported [26]. When cells grew logarithmically, the cells were seeded in 96-well plates (2 × 10^5^ cells/well, 100 μL/well). After incubation for 24 h, we aspirated the original medium and added a new 200 μL medium with/without the various concentrations of compounds (3.125–50 μM) or l-NNMMA (3.125–50 μM)/LPS (1 μg/mL) treatment. After incubation for 24 h again, 50 μL of the supernatant of each well was transferred to another 96-well plate and 50 μL of NO kit solution (Beyotime, Shanghai, China) Ι (Sulfanilamide) and Π (N-(1-naphthyl) ethylenediamine dihydrochloride) were added sequentially. Then, absorbance was measured at 540 nm using a microplate reader. The positive control used was Quercetin. The formula (NO inhibition rate = [(ODLPS − ODCon) − (ODcompound-ODCon)]/(ODLPS − ODCon) × 100%) was used to calculate the anti-inflammatory activities.

#### 2.4.4. Western Blotting

The RAW 264.7 cells were plated in 6-well culture plates (Flat bottom, Labselect, Hefei, Anhui, China, 2 mL/well) at a density of 8 × 10^6^ cells per well. Total cellular proteins were extracted using 150 μL of freshly prepared RIPA lysis buffer prepared by mixing PMSF, phosphatase inhibitor cocktail, and RIPA buffer in a 1:1:100 ratio (Beyotime, Shanghai, China). Protein samples were electrophoretically separated on a 10% sodium dodecyl sulfate-polyacrylamide gel (SDS-PAGE, Epizyme, Shanghai, China) and subsequently transferred to a polyvinylidene difluoride (PVDF, Millipore, Bedfoerd, MA, USA) membrane via electroblotting. The membrane was subjected to a 15 min blocking step with rapid blocking buffer (Beyotime, Shanghai, China) prior to overnight incubation with primary antibodies (Huabio, Hangzhou, Zhejiang, China) at 4 °C. Following three 10 min washes with Tris-buffered saline containing 0.1% Tween-20 (TBST, Epizyme, Shanghai, China), the membrane was probed with horseradish peroxidase (HRP)-conjugated secondary antibodies (Huabio, Hangzhou, Zhejiang, China) for 1 h at room temperature. Protein bands were detected using enhanced chemiluminescence (ECL, Glpbio, Montclair, CA, USA) substrate with a 1 min exposure time, and visualized through a gel documentation system (Bio-rad, Heracles, CA, USA).

#### 2.4.5. Statistical Analysis

A one-way analysis of variance (ANOVA) was conducted to examine the differences in cell viability in the compound group and normal group. The dose–response relationship was analyzed through four-parameter logistic nonlinear regression modeling to calculate the half-maximal inhibitory concentration (IC_50_) with 95% confidence intervals. The statistical analysis was performed using the GraphPad 10 software. All the data were presented as mean ± standard deviation (SD) of at least three biological replicates.

### 2.5. ECD Calculation and ^13^C NMR Calculation

The procedures were the same as previously described [27]. For **4**, the geometric optimization was performed at the gas phase at the B3LYP/6-31+G(d) level. Then, the ECD calculation was executed using the time-dependent density functional theory (TDDFT) methodology [28,29] at the rB3LYP/6-311G level.

Merck Molecular Field in Spartan’s 10 software was used for the conformational analysis of **3**. Conformers with populations exceeding 5% according to the Boltzmann distribution were optimized using the B3LYP/6-31G (d, p) level in the polarizable continuum model (PCM) with methanol as the solvent (Gaussian 09). Subsequently, NMR calculations were performed using the gauge invariant atomic orbital (GIAO) method at the mPW1PW91-SCRF/6-31G (d, p) level with PCM in methanol (Gaussian 05). Finally, the shielding constants were averaged using the Boltzmann distribution theory for each stereoisomer, and their experimental and calculated data were analyzed using DP4+ probability.

## 3. Results

### 3.1. Structure Elucidation

Dothiofuran A (**1**), isolated as a white solid, had a molecular formula of C_15_H_14_O_5_ with nine degrees of unsaturation as determined by HRESIMS. The ^1^H NMR spectrum revealed three methyl groups at δ_H_ 2.09 (s, H_3_-13), 2.18 (s, H_3_-10), 2.32 (s, H_3_-12), two aromatic proton signals at δ_H_ 7.11 (d, *J* = 7.8 Hz, H-4), 7.76 (d, *J* = 7.8 Hz, H-5), and one methylene group at δ_H_ 4.16 (s, H-6). The ^13^C NMR and HSQC spectral displayed 15 carbon signals, including three methyl groups, one methylene, two methines, and nine non-hydrogenated carbons (two carbonyl groups and a carboxyl group). The NMR data of **1** (Table 1) showed great similarity to those of **8,** indicating that **1** was a benzofuran derivative. The deduction was supported by the HMBC cross-peaks (Figure 2) from H-4 to C-3, C-3a, and C-9b; from H-5 to C-5a; from H-6 to C-5a and C-9a; from H-12 to C-2 and C-11; and from H-13 to C-11. Moreover, the HMBC cross-peaks (Figure 2) from H-6 to C-5a, C-7, C-9a, and C-9, from H_3_-10 to C-7, indicated that one carboxyl group was located at C-9a, and one acetonyl group was replaced at C-5a. Thus, the structure of **1** was established as shown in Figure 1.

Dothiofuran B (**2**) was obtained as a yellow oil, and has the same molecular formula as **1**. The ^1^H and ^13^C NMR data (Table 1) were similar to those of **1**. The HMBC cross-peaks (Figure 2) from H-4 to C-3, C-3a, and C-9b; from H-5 to C-5a; from H-6 to C-5a and C-9a; from H-12 to C-2 and C-11; and from H-13 to C-11 indicated that **2** was a benzofuran derivative. A comparison of the 1D NMR data of **2** and **8** suggested that **2** was the oxidation product of **8**. The conclusion was supported by the HMBC correlations (Figure 2) from H-6 to C-9. Notably, both the optical rotation and CD maximum of **2** were close to zero. According to the hemiacetal structure at C-7, compound **2** could not be separated by a chiral column.

Dothiofuran C (**3**) was obtained as a white solid. Its molecular formula was assigned as C_15_H_20_O_4_ by the HRESIMS. A comparison of the NMR data (Table 1) of compounds **3** and **1** revealed that **3** was a derivative of **1**. The HMBC correlation from H_3_-12 to C-2 and C-11 and from H-4 to C-3 and C-3a indicated that the Δ^2(11)^ double bond and the ketocarbonyl group at C-3 in **1** have disappeared, and a hydroxy group has replaced at C-11 in **3**. The HMBC cross-peaks (Figure 2) from H-6 to C-5a and C-9a and from H2-9 to C-5a, C-9a, and C-9b revealed that the carboxyl group at C-9a was reduced to an alcohol group, and a hydroxy group has located at C-6. In addition, the HMBC cross-peak from H-7 to C-6 suggested that a terminal olefin fragment was substituted at C-6. Thus, compound **3** was elucidated as shown in Figure 1. Considering the absence of the correlation from H-2 to H-6 in the NOESY spectrum, the ^13^C NMR calculations of (2*R**, 6*R**)-**3**a and (2*R**, 6*S**)-**3**b were performed using the GIAO method at mPW1PW91-SCRF/6–31 G (d, p)/PCM (Chloroform) (Appendix A). Ultimately, conducting a DP4+ probability analysis led to the conclusion that the most probable structure for **3** was 2*R**, 6*S**, which was supported by a better correlation coefficient (R^2^ = 0.9996) and a high DP4+ probability of 100% (all data) (Figure 3). Due to a lack of sufficient quantity, the absolute configuration of **3** has not been determined.

Dothiopene A (**4**) was obtained as a colorless oil, and had the molecular formula C_15_H_22_O_3_ based on the HRESIMS data, with five degrees of unsaturation. The ^1^H NMR spectrum showed four methyl groups at δ_H_1.04 (s, H_3_-9′), 1.08 (s, H_3_-8′), 1.99 (s, H_3_-7′), and 2.18 (s, H_3_-6), and two olefinic proton signals at δ_H_ 5.86 (s, H-5′) and 5.71 (s, H-2). The ^13^C NMR displayed signals for 15 carbon signals, including four methyl groups, three methylenes, three methines, and five non-hydrogenated carbons. Its ^1^H and ^13^C NMR data (Table 1) showed a close resemblance to those of (10*S*, 2*Z*)-3-methyl-5-(2,6,6-trimethyl-4-oxocyclohex-2-enyl) pent-2-enoicacid [30]. An analysis of the ^1^H−^1^H COSY correlations revealed one spin coupling system H-4/H-5/H-1′. The HMBC cross-peaks from H-1′ to C-2′ and C-6′, from H_3_-7′ to C-5′ and C-6′, from H-3′ and H-5′ to C-4′, from H_3_-8′ to C-2′ and C-3′, and from H_3_-9′ to C-2′ indicated the presence of the cyclohexanone segment, and that the spin coupling system has been replaced at C-1′. In addition, the HMBC cross-peaks from H-6 to C-2, C-3, and C-4 and from H-2 to C-1 showed that the acrylic acid moiety was located at C-4, and one methyl group was substituted at C-3. Thus, the structure of **3** was established as shown in Figure 1. The double bond at C-2 and C-3 was determined as a trans configuration by the NOESY correlation of H-2/H_3_-6 (Figure 4). The absolute configuration at C-1′ was determined as 1′*R* by ECD calculation (Figure 5).

Phomosterol C (**5**) was isolated as a white solid. The molecular formula was established as C_27_H_40_O_4_ from the HRESIMS data, indicating 8 degrees of unsaturation. The ^1^H NMR spectrum (Table 2), showed the presence of two aromatic protons at δ_H_ 6.90 (d, *J* = 7.9 Hz), δ_H_ 6.92 (d, *J* = 7.9 Hz), and four methyl groups at δ_H_ 0.60 (s), 0.81 (d, *J* = 6.6 Hz), 0.93 (d, *J* = 6.9 Hz) and 0.95 (d, *J* = 6.6 Hz). The ^13^C NMR data (Table 2) displayed 27 carbon signals ascribed to four methyls, eight methylenes (one oxygenated), nine methines (two olefinic and two oxygenated), and six nonprotonated carbons (one carbonyl and four olefinic). After comparing the NMR data (Table 1) of **5** and **6**, we concluded that **5** is a 1-hydroxylated derivative of **6**. The ^1^H-^1^H COSY correlation of H-1/H_2_-2/H-3/H_2_-4 along with HMBC cross-peaks from H-1 to C-8 and C-9 (Figure 2) further proved the conclusion above. The relative configuration of compound **5** was elucidated by analyzing the NOESY correlations (Figure 4). The correlations of H-1/H-11a/H_3_-18 (/H-20)/15*β* indicated that those protons were on the same side. The cross-peaks of H-14/H-17/15*a* suggested that these protons were on the other side. The absolute configuration of **5** was determined by a similar experimental ECD curve (Figure 6) of **5** to that of **7**.

### 3.2. Anti-Inflammatory Screening of Extracts

In vitro, the inflammatory model was established by stimulating the RAW264.7 macrophage cells with 1 μg/mL LPS to elevate nitric oxide (NO) production. The NO inhibitory activity of all the isolated compounds was investigated to screen potential anti-inflammatory candidates. NG-Monomethyl-l-arginine, Monoacetate Salt (l-NMMA) has been used as a paradigmatic inhibitor of NO synthase as a positive inhibitor (IC_50_, 17.7 μM). Among the isolates, **5** and **7** showed significant NO inhibitory effect in LPS-induced RAW264.7 murine macrophages (Figure 7A,B) without any significant impact on the cell viability of RAW264.7 macrophage cells at the concentration of 3.125–12.5 μM (Figure 7C,D and Appendix A). As a result, **7** exhibited significant anti-inflammatory activity with an IC_50_ value of 4.6 μM. Compound **5** showed moderate activity with an IC_50_ value of 29.3 μM. The other compounds have not exhibited potent inhibitory activities at the concentration of 50.0 μM. All the compounds showed no cytotoxic effect at the tested concentration.

The inducible nitric oxide synthase (iNOS) and cyclooxygenase-2 (COX-2) play pivotal roles in mediating inflammatory responses [31]. In this study, the effect of **7** on the translational upregulation of iNOS and COX-2 in LPS-stimulated RAW 264.7 cells was evaluated. As demonstrated in Figure 8, compared with the untreated control group, LPS stimulation significantly upregulated both iNOS and COX-2 protein levels. Notably, **7** exhibited a dose-dependent and statistically significant suppression of iNOS overexpression but it did not show a substantial inhibitory effect on COX-2 expression across the tested concentrations. This selective inhibition pattern suggests that **7** specifically targets the iNOS signaling pathway rather than the COX-2-mediated inflammatory cascade.

## 4. Conclusions

In this article, one strain *Dothiorella* sp. ZJQQYZ-1 from *Kandelia candel* was studied. As a result, three new polyketides dothiofurans A-C (**1**–**3**), one new sesquiterpene dothiopene A (**4**), one new steroid phomosterol C (**5**), and three known analogs (**6**–**8**) were isolated. Their structures of new compounds were determined by NMR spectroscopy, HRESIMS, ECD, and ^13^C NMR calculations. All the compounds were evaluated for their inhibitory activities on NO production in LPS-induced RAW264.7 macrophages. Compound **7** exhibited significant anti-inflammatory activity with an IC_50_ value of 4.6 μM, which was better than the positive control l-NMMA. Moreover, **7** exerted anti-inflammatory effects by inhibiting the iNOS pathway. Compound **7** was characterized by an unusual aromatic B ring skeleton in sterols. Those compounds were reported to have antibacterial, anti-HIV, anti-inflammatory, and neuroprotective activities [23,32,33,34]. This study suggested that the phytosterols could be further researched as anti-inflammatory therapeutic lead compounds.

## Figures and Tables

**Figure 1 microorganisms-13-00890-f001:**
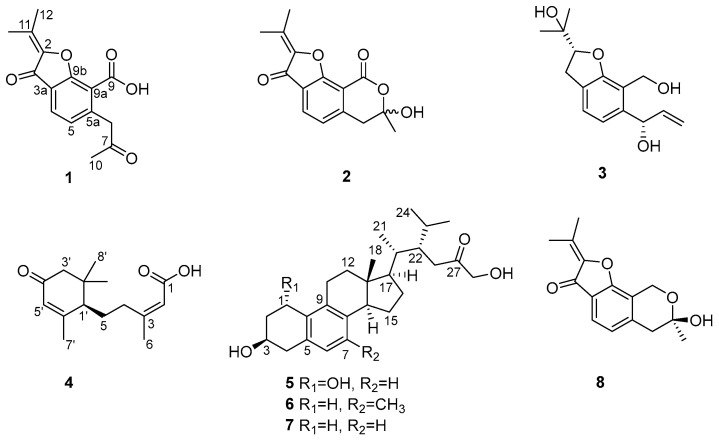
Chemical structures of **1**–**8**.

**Figure 2 microorganisms-13-00890-f002:**
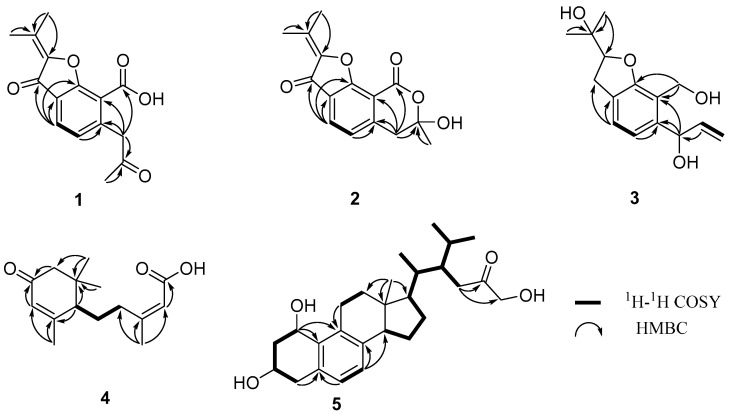
The key correlations of ^1^H-^1^H COSY and HMBC of **1**–**5**.

**Figure 3 microorganisms-13-00890-f003:**
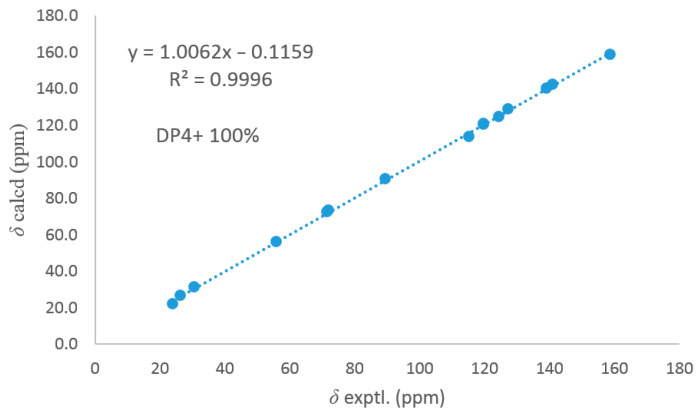
Comparisons of calculated and experimental ^13^C NMR data of **3**.

**Figure 4 microorganisms-13-00890-f004:**
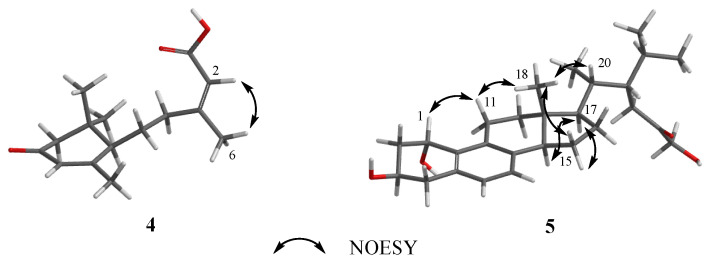
Key NOESY correlations of **4** and **5**.

**Figure 5 microorganisms-13-00890-f005:**
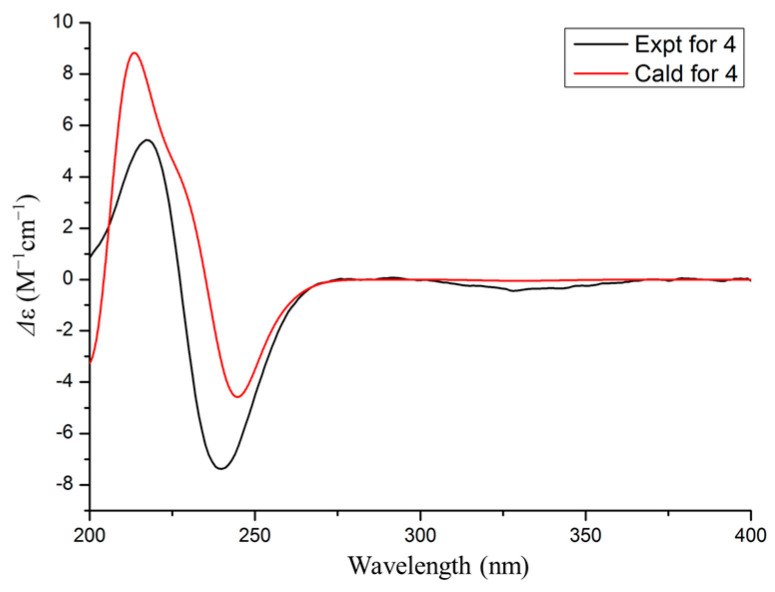
Experimental and calculated ECD spectra of **4** in MeOH.

**Figure 6 microorganisms-13-00890-f006:**
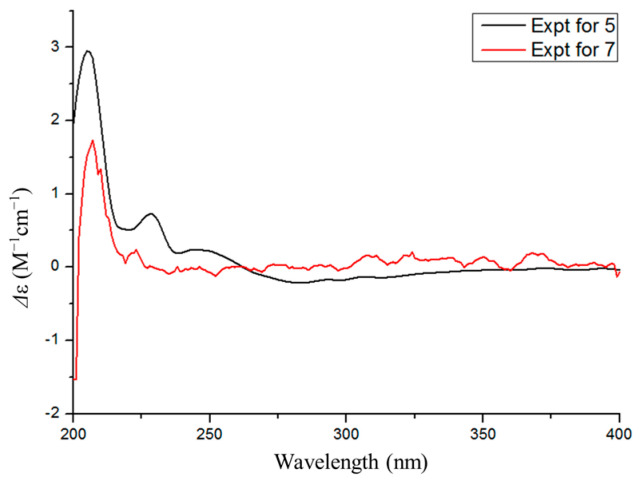
Experimental ECD spectra of **5** and **7**.

**Figure 7 microorganisms-13-00890-f007:**
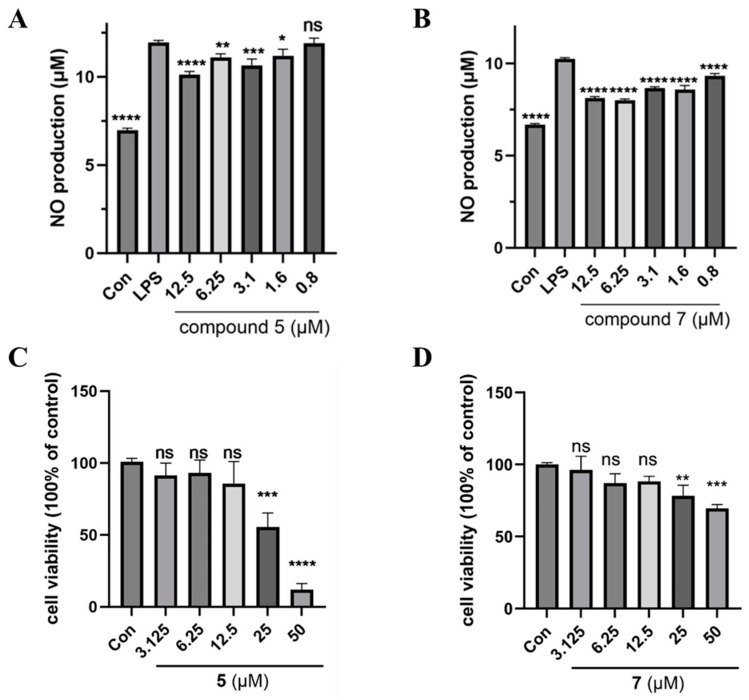
The inflammatory effects of **5** and **7** on LPS (1 μg/mL)-induced NO production in RAW264.7. The NO levels (**A**,**B**) and cell viabilities (**C**,**D**) were measured using Griess reagent and CCK8 reagent, respectively. Data were expressed as mean values ± SD, *n* = 3. * *p* < 0.05, ** *p* < 0.01, *** *p* < 0.001, and **** *p* < 0.0001.

**Figure 8 microorganisms-13-00890-f008:**
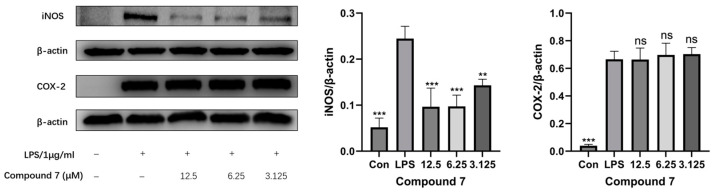
Protein levels of COX-2, iNOS, and *β*-actin by Western blot of **7**. All the presented data are expressed as the mean ± SD, with *n* = 3. ns, not significant; ** *p* < 0.01, and *** *p* < 0.001 versus the LPS group.

**Table 1 microorganisms-13-00890-t001:** ^1^H (500 MHz) and ^13^C (125 MHz) NMR data for **1**−**4**.

	1		2		3			4	
No.	*δ*_H_ *^a^* (*J* in Hz)	*δ*_C_ *^a^*	*δ*_H_ *^b^* (*J* in Hz)	*δ*_C_ *^b^*	*δ*_H_ *^c^* (*J* in Hz)	*δ*_C_ *^c^*	No.	*δ*_H_ *^c^* (*J* in Hz)	*δ*_C_ *^c^*
2		144.5		145.2	4.58, t (8.9)	89.4	1		170.3
3*α*		182.3		182.5	3.10, dd (9.5, 15.7)	30.5	2	5.71, s	115.3
3*β*					3.20, dd (8.4, 15.7)		3		161.8
3a		123.0		123.9		127.2	4	2.26, m	40.7
4	7.77, d (7.8)	126.2	7.88, d (7.8)	129.5	7.05, d (7.5)	124.4	5*α*	1.58, m	28.0
5	7.11, d (7.8)	127.1	7.09, d (7.8)	122.4	6.84, d (7.5)	119.8	5*β*	1.88, m	
5a		118.4		109.9		140.9	6	2.18, s	19.2
6	4.16, s	49.1	3.26, d (16.9)	39.4	5.39, s	72.0	1′	1.89, m	50.6
			3.37, d (16.9)				2′		36.3
7		204.7		104.9	6.12, m	139.0	3′*α*	2.07, d (17.4)	47.0
9*α*		166.0		163.9	4.65, d (12.0)	55.8	3′*β*	2.37, d (17.4)	
9*β*					4.78, d (12.0)		4′		199.1
9a		145.2		147.6		119.7	5′	5.86, s	125.4
9b		162.3		161.0		158.7	6′		164.5
10*α*	2.18, s	30.3	1.70, s	22.0	5.26, d (10.5)	115.2	7′	2.0, s	24.6
10*β*					5.37, d (9.1)		8′	1.08, s	27.1
11		133.5		135.8		71.6	9’	1.04, s	28.7
12	2.32, s	17.3	2.37, s	17.0	1.15, s	23.9			
13	2.09, s	20.3	2.22, s	20.0	1.36, s	26.5			

*^a^* measured in DMSO-*d*_6_. *^b^* measured in CDCl_3_ and CD_3_OD. *^c^* measured in CDCl_3_.

**Table 2 microorganisms-13-00890-t002:** ^1^H (500 MHz) and ^13^C (125 MHz) NMR data for compound **5** in CD_3_OD.

	5			5	
No.	*δ*_H_ (*J* in Hz)	*δ* _C_	No.	*δ*_H_ (*J* in Hz)	*δ* _C_
1	5.04, m	65.2	14	2.68, dd (8.0, 11.3)	51.5
2*α*	1.74, m	41.4	15*α*	2.08, m	23.6
2*β*	2.29, m		15*β*	1.56, m	
3	4.26, m	62.5	16*α*	2.15, m	28.5
4*α*	3.11, dd (4.8, 15.3)	38.9	16*β*	1.49, m	
4*β*	2.59, dd (10.8, 15.3)		17	1.42, m	53.1
5		133.3	18	0.60, s	10.6
6	6.92, d (7.9)	125.4	20	1.88, m	36.1
7	6.90, d (7.9)	126.3	21	0.93, d (6.7)	20.0
8		138.2	22	2.02, m	41.3
9		135.5	23	1.55, m	30.9
10		132.3	24	0.95, d (6.6)	12.7
11*a*	2.94, m	23.1	25	0.81, d (6.6)	20.7
11*β*	3.17, dd (7.2, 18.0)		26*α*	2.40, m	36.8
12*α*	2.19, m	36.8	27		211.0
12*β*	1.75, m		28	4.27, m	67.4
13		42.0			

## Data Availability

The original contributions presented in this study are included in the article/Appendix A. Further inquiries can be directed to the corresponding authors.

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
