# Peer review of "Metabolites with Anti-Inflammatory Activities Isolated from the Mangrove Endophytic Fungus Dothiorella sp. ZJQQYZ-1"

_microorganisms, 2025, doi:10.3390/microorganisms13040890_

Round 1
Reviewer 1 Report (Previous Reviewer 1)
Comments and Suggestions for Authors
Once a few trivial points are addressed, the manuscript could be acceptable for publication in Microorganisms.
- At the beginning of a sentence, compound numbers should be indicated with "Compound." However, within the sentence, it is not necessary to use "compound" before the numbers.
- The reviewer indicated that the location of the company from which the reagents and instruments were purchased, specifying the city, state, and country in the Experimental section should be presented. According to my indication, the authors described “Thermo Fisher, Waltham, MA, USA” on line 59. However, when it appears on the second day, it is sufficient to mention only the company name. On line 132, “Thermo Fisher, Waltham, MA, USA” should be revised to “Thermo Fisher”. The same applies to others.
- On line 197: “Thus, the planar structure of compound 1 was established.” should be revised to “Thus, the structure of 1 was established as shown Figure 1”. Since 1 does not have a chiral carbon atom, "planar" is not necessary. Also, please specify that the structure of 1 is shown in Figure 1. The same applies to others.
Author Response
- At the beginning of a sentence, compound numbers should be indicated with "Compound." However, within the sentence, it is not necessary to use "compound" before the numbers.
Response: Thanks for your suggestion. Numbers at the beginning of the sentence have deleted “compound” in the manuscript.
- The reviewer indicated that the location of the company from which the reagents and instruments were purchased, specifying the city, state, and country in the Experimental section should be presented. According to my indication, the authors described “Thermo Fisher, Waltham, MA, USA” on line 59. However, when it appears on the second day, it is sufficient to mention only the company name. On line 132, “Thermo Fisher, Waltham, MA, USA” should be revised to “Thermo Fisher”. The same applies to others.
Response: Thanks for your suggestion. The mistakes have revised in the manuscript.
- On line 197: “Thus, the planar structure of compound 1 was established.” should be revised to “Thus, the structure of 1 was established as shown Figure 1”. Since 1 does not have a chiral carbon atom, "planar" is not necessary. Also, please specify that the structure of 1 is shown in Figure 1. The same applies to others.
Response: Thanks for your suggestion. The mistake has revised in the manuscript.
Reviewer 2 Report (Previous Reviewer 3)
Comments and Suggestions for Authors
Thank you for submitting the revised and well improved manuscripts. All the previous comments were addressed. However I have minor comments:
- In line 61, line spacing is required.
- Section 2.4.3 the title need to be revised e.g In vitro measurement of Nitric oxide production.
- figure 3: The resolution need to be improved. it is not clear
- Section 3.2 Provide the name of the cells that were used in anti-inflammatory assay
- Provide the name/ specification of the 96 well plate that were used. Is it a round bottom or flat bottom? Name of the company and the country.
- During 24 hour incubation in both cytotoxicity and anti-inflamatory: provide the conditions eg temperature and carbon dioxide level
- Name and the origin of the cells used for Western blot? Name of the 6 well plate.
Author Response
- In line 61, line spacing is required.
Response: Thanks for your suggestion. The mistake has revised in the manuscript.
- Section 2.4.3 the title need to be revised e.g. In vitro measurement of Nitric oxide production.
Response: Thanks for your suggestion. The title has revised in the manuscript.
- figure 3: The resolution need to be improved. it is not clear
Response: Thanks for your suggestion. The “Figure 3” has improved in the manuscript.
- Section 3.2 Provide the name of the cells that were used in anti-inflammatory assay.
Response: Thanks for your suggestion. RAW264.7 macrophage cells were used in anti-inflammatory assay. It has revised in the manuscript as follows:
“In vitro, inflammatory model was established by stimulating the RAW264.7 macro-phage cells with 1 μg/ml LPS to elevate nitric oxide (NO) production”
- Provide the name/ specification of the 96 well plate that were used. Is it a round bottom or flat bottom? Name of the company and the country.
Response: Thanks for your suggestion. The mistakes have revised in the manuscript as follow:
“In a nutshell, when cells grow logarithmically, cells were seeded in 96-well plates (Flat bottom, Labselect, Hefei, Anhui, China, 2 × 105 cells/well, 100 μL/well)”
- During 24 hour incubation in both cytotoxicity and anti-inflamatory: provide the conditions eg temperature and carbon dioxide level
Response: Thanks for your suggestion. The mistakes have revised in the manuscript as follow:
“After incubation for 24 hours (5% CO2, 37 °C)”
- Name and the origin of the cells used for Western blot? Name of the 6 well plate.
Response: Thanks for your suggestion. The mistakes have revised in the manuscript as follow:
“The RAW 264.7 cells were plated in 6-well culture plates (Flat bottom, Labselect, Hefei, Anhui, China, 2 mL/well) at a density of 8 × 10⁶ cells per well.”
This manuscript is a resubmission of an earlier submission. The following is a list of the peer review reports and author responses from that submission.
Round 1
Reviewer 1 Report
Comments and Suggestions for Authors
Please refer to the attached sheet

There are numerous issues with English grammar and word choice.
Reviewer 2 Report
Comments and Suggestions for Authors
The Abstract lacks of anti-inflammatory methodology, general conclusions and recommendations.
The introduction states “Currently, anti-inflammatory drugs commonly used in clinical practice include 19 glucocorticoids and non-steroidal drugs, but all of them have side effects, such as drug resistance and allergic reactions” How do we know that the proposed compounds will be free of side effects?.
In the methodology cite permission for the collection of Kandelia candel.
In order to make the experiments reproducible anywhere in the world, it is suggested to report an average temperature instead of room temperature.
DMEM is not Eagle Medium.
What is CCK8 solution? as well as NO kit solution Ι and Π ?
The quercetin used as a positive control, at what concentrations was it used?
What is the TDDFT methodology?
Figure 4 is presented in the manuscript first than Figure 3.
In Figure 6 in panels C and D the statistical analyses are not presented. Additionally, the legend of this figure does not mention the statistical tools to determine the differences. What is the control in panels C and D, the quercetine?. What was the LPS concentration, cite this in the figure 6 legend.
It is recommended to perform an analysis of results in a new item or together with the results of but in a more in-depth way relating structure to bioactivity by making comparisons with other studies. The analysis should be deeper and broader, not limited to describing the elucidation of chemical structures.
Regarding the decrease in NO production, Figure 6 shows a greater effect with compound 7 than with compound 5. However, compound 5 has a lower IC50 than compound 7, how can this result be explained?
Compounds that decrease NO levels in inflamed macrophages have a high anti-inflammatory potential. However, further studies are essential to fully understand their mechanism of action and safety profile.
The conclusion is a synthesis of the results and should be restated.
Reviewer 3 Report
Comments and Suggestions for Authors
Reviewers comments
Abstract
- There is no information on the background of the study and the aim of the study. What is the problem statement of this study? Material and methods- This section in the abstract is not clearly explained. The results and discussion are poorly written.
- The abstract needs major revision so that it can be easy to understand the overall study
- Material and methods section
- The order of experimental procedure is incoherent; I suggest that the fungal isolation be the first to be written, and the chemistry part is at the end.
- Cell culture- Define the origin of the cells RAW264.7 macrophage cell. Where were they isolated from?
- Cell viability assays
- How did you calculate the percentage of cell viability? Please provide the formula that you have used to calculate the %cell viability
- What positive and negative controls were used in this assay?
- Provide the source of the CCK8 solution and how it works. Describe the principle of CCK8
- What is the volume of the media you added to each well and the volume of the cells added?
- NO production
- What was the volume of cells added to each well and the volume of the media?
- What was the concentration of compound added in each well?
- What was the concentration of compound added in each well?
- How did you calculate the anti-inflammatory activities? Provide the formula used
- Results
- Figures and table legends need to be revised, and more information needs to be provided.
Table 1, Figures 1,2,3 and 4
- Figures A and B are not adequately discussed and explained in the text.
- How did you calculate the IC50? What is the accepted (standard) IC50 required for the anti-inflammatory?
- Discussion and conclusion
- This section is not well written. The results were not discussed well.
- Major revision.
Reviewer 4 Report
Comments and Suggestions for Authors
- Line 26: Specify the types of saccharides involved.
- Lines 35-37: Clarify whose attention is being referred to in this section.
- Lines 38-41: The introduction section should provide background information on the fungus, related species within the same genus, and relevant context, rather than describing the results as seen in lines 35-45.
- Section 2.1: Please include references to relevant literature. Additionally, provide a more detailed description of the methodology and experimental conditions to ensure reproducibility; this information could be added to the supplementary material.
- Line 64: Clarify the title—specifically, what was fermented, extracted, and isolated.
- Tables: Ensure the tables are formatted correctly and their titles are clear and descriptive.
- Results Section: The results section is inadequately described and discussed. Please provide more detailed comparisons with similar fungi and compounds. Consider a deeper discussion of the anti-inflammatory potential based on the observed results, referencing similar studies.
The manuscript, in its current form, is difficult to read and lacks fluidity in its discourse. In certain sections, the descriptions and discussions are insufficient. It is recommended to rephrase sentences for clarity and provide more detailed explanations to enhance the communication and interpretation of the data.